# XRCC1 Prevents Replication Fork Instability during Misincorporation of the DNA Demethylation Bases 5-Hydroxymethyl-2′-Deoxycytidine and 5-Hydroxymethyl-2′-Deoxyuridine

**DOI:** 10.3390/ijms23020893

**Published:** 2022-01-14

**Authors:** María José Peña-Gómez, Marina Suárez-Pizarro, Iván V. Rosado

**Affiliations:** Centro Andaluz de Biología Molecular y Medicina Regenerativa (CABIMER), Universidad de Sevilla-CSIC-Universidad Pablo de Olavide, 41092 Seville, Spain; marpengom7@gmail.com (M.J.P.-G.); suarezmarina99@gmail.com (M.S.-P.)

**Keywords:** XRCC1, replication fork instability by 5hmC, 5hmU-mediated genomic instability, epigenetic DNA bases

## Abstract

Whilst avoidance of chemical modifications of DNA bases is essential to maintain genome stability, during evolution eukaryotic cells have evolved a chemically reversible modification of the cytosine base. These dynamic methylation and demethylation reactions on carbon-5 of cytosine regulate several cellular and developmental processes such as embryonic stem cell pluripotency, cell identity, differentiation or tumourgenesis. Whereas these physiological processes are well characterized, very little is known about the toxicity of these cytosine analogues when they incorporate during replication. Here, we report a role of the base excision repair factor XRCC1 in protecting replication fork upon incorporation of 5-hydroxymethyl-2′-deoxycytosine (5hmC) and its deamination product 5-hydroxymethyl-2′-deoxyuridine (5hmU) during DNA synthesis. In the absence of XRCC1, 5hmC exposure leads to increased genomic instability, replication fork impairment and cell lethality. Moreover, the 5hmC deamination product 5hmU recapitulated the genomic instability phenotypes observed by 5hmC exposure, suggesting that 5hmU accounts for the observed by 5hmC exposure. Remarkably, 5hmC-dependent genomic instability and replication fork impairment seen in *Xrcc1*^−/−^ cells were exacerbated by the trapping of Parp1 on chromatin, indicating that XRCC1 maintains replication fork stability during processing of 5hmC and 5hmU by the base excision repair pathway. Our findings uncover natural epigenetic DNA bases 5hmC and 5hmU as genotoxic nucleosides that threaten replication dynamics and genome integrity in the absence of XRCC1.

## 1. Introduction

The genetic information encoded in the genome is constantly under endogenous and exogenous threat [1]. During evolution, cells evolved efficient DNA repair processes to rapidly correct these predominant forms of DNA damage. One of the most common types of DNA injuries is nucleobase damage, comprising oxidation, deamination and alkylation. Many base lesions are pro-mutagenic (i.e., 7,8-dihydro-8-oxoguanine, 8-oxoG), as their chemical structures favour non-canonical base-pairing, thus giving rise to transition or transversion mutations [2]. Likewise, cytosine and 5-methylcytosine (5mC) modifications—either by Ten-Eleven Translocation (TET)-mediated catalytic oxidation or by hydrolytic deamination of the bases during the cytosine demethylation process [3]—give rise to a plethora of cytidine and uridine analogues, whose effects on the mutational status and their physiological consequences could be potentially devastating for cell survival. These small nucleobase modifications are mainly detected and repaired by the base excision repair (BER) system [4,5]. BER is initiated by one of eleven DNA glycosylases, which efficiently function on one hand as sensors of the type of modification and on the other hand as executors of the non-distorting damaged DNA base by performing the glycosyl bond cleavage step [6]. The type of damaged nucleobase and the DNA glycosylase employed dictate the chemical structure of the single strand break (SSB) intermediate created during the fixation process. Upon incision of the sugar-phosphate backbone by the AP endonuclease 1 (APE1), the XRCC1 scaffold protein serves as a platform for the coordinated recruitment of polymerase β (POLβ), which fills the one-nucleotide or larger gap, by either Ligase IIIα (LIG3), the enzyme of choice to carry the ligation step during short-patch BER, or Ligase 1 (LIG1) during the long-patch subpathway [4,6,7]. Owing to the fact that BER intermediates are passed on from one enzyme to the next, enabling the so-called “handover model”, which has been compared to “passing the baton” in a relay race, BER is considered a highly coordinated and efficient pathway of repair base damage [8]. Despite the “handover” model, under some conditions of “unscheduled” SSBs or stochastic core BER components disassembly during intermediate processes, PARP1 and PARP2 are recruited to these incomplete intermediates to enhance the recruitment of the XRCC1 preligation complex in order to accelerate the repair reaction [9]. XRCC1 is a scaffold protein comprised of an N-terminal domain that mainly interacts with POLβ, a central BRCT domain that binds DNA and PARP1, and a C-terminal BRCT domain that interacts with LIG3 [10,11]. Upon cell exposure to the broad range cell cycle-independent alkylating agent MMS, PARP1 is activated and required to recruit XRCC1 at SSBs to promote its accumulation [12]. 

In addition to its well-established role during SSB repair and BER, XRCC1 also carries out other functions associated to the maintenance of the replisome stability [13,14]. At the replication fork, XRCC1 has been found to physically interact with several bona fide replication factors (i.e., PCNA, DNA polymerase α-primase, FEN1) [15,16,17]. During ligation of Okazaki fragments processing (OFP), PARP1 promotes recruitment of XRCC1 to accelerate nick ligation, whereas the use of PARP inhibitors (PARPi) disrupts OFP ligation, leaving unrepaired SSBs, which are further converted into double strand breaks (DSBs) during DNA replication [17,18]. In agreement with this notion, mutations in the XRCC1 gene mainly associate with neurological disorders occurring by excessive accumulation of unrepaired SSB in postreplicative tissues [5,10], and cancer predisposition, presumably as a consequence of increased replication-associated DSBs formation [13,14]. XRCC1 has also been reported to play crucial roles during microhomology-mediated end joining (MMEJ) and replication fork restart in the absence of the fork protector BRCA2, by promoting nascent DNA degradation mediated by MRE11 and mutagenic repair [14]. Since the discovery that PARPi sensitize homologous recombination (HR)-deficient tumours, the primary sensitizing lesion has been attributed to the formation of toxic DSBs during replication fork collapse [19,20] and, more recently, to lagging strand-associated single stranded gaps [21]. Olaparib traps PARP1 on DNA, interferes with DNA replication and promotes fork collapse, necessitating a proficient HR mechanism for full restoration of an intact fork structure [22,23,24].

Here, we describe a protective role for the BER factor XRCC1 during misincorporation of cytidine analogues. Both 5hmC and 5-formyl-2′-deoxycytosine (5fC), but not 5mC or 5-carboxy-2′-deoxycytosine (5caC), activated the BER pathway during their misincorporation. Moreover, *Xrcc1* deficient cells displayed marked sensitivity upon 5hmC or 5hmU exposure, a 5hmC deamination product. 5hmC or 5hmU exposure led to increased chromatin retention of PARP1 and replication fork instability, that were exacerbated by treatment with the PARP1-trapping drug olaparib. Remarkably, replication fork instability by 5hmC occurred during the ongoing S-phase, thus suggesting a novel role of XRCC1 at coordinating BER during replication fork dynamics. Collectively, these data suggest that persistent PARP1 at BER intermediate structures during removal of 5hmC-derived lesions are a novel source of replication fork insults that challenge genome integrity.

## 2. Results

### 2.1. 5hmC or 5fC Cytidine Analogues Elicits a BER Response in the Human U2OS Cell

During the active removal of 5fC and 5caC, BER has been proposed as universal mechanism to repair AP endonuclease-mediated SSBs that arise from Thymine-DNA glycosylase (TDG)-dependent AP sites. Upon base removal by TDG, and under certain conditions PARP1 PARylates itself and many other proteins to recruit BER factors to the SSB. This event, in turn, recruits the XRCC1/LIG3α/POLβ preligation complex to seal the nick, thus restoring the DNA strand (Figure 1A). Whereas the role of BER in removing epigenetically modified cytidine analogues from the template DNA is well understood, little is known about its role during misincorporation of 2-deoxycytidine analogues 5mC, 5hmC, 5fC and 5caC during replication. To evaluate whether BER pathway is activated by misincorporation of these cytidine analogues during DNA synthesis in an unbiased manner, we generated a U2OS reporter cell line stably expressing an RFP-tagged XRCC1 construct (Figure 1B). Upon selection with G418 and FACS sort RFP+ cells to almost 95% RFP+ cell population, RFP-tagged XRCC1 U2OS cells showed a dim cytoplasmic fluorescence and a brighter nuclear signal (Figure 1B). We then examined the recruitment of XRCC1-RFP to chromatin by an immunofluorescence-based chromatin retention assay, whereby the cell permeabilization step precedes the formaldehyde fixation, allowing the detection of only tightly bound chromatin factors. As a control of the technique, U2OS +XRCC1-RFP cells exposed to hydrogen peroxide (H_2_O_2_), a well-known oxidative agent that strongly induces base oxidation and BER activation, promoted chromatin retention of XRCC1-RFP on chromatin which is completely absent in the untreated condition (Figure 1B), thus validating our U2OS reporter cell line. We next examined whether incorporation of any of the cytosine analogues normally present in the genome promoted XRCC1-RFP chromatin retention when supplemented exogenously. Unlike dC, 5mC or 5caC, 5hmC and 5fC consistently induced XRCC1-RFP chromatin retention in our settings (Figure 1C). Moreover, 5hmC and 5fC exposure led to heightened DSB formation measured as increased in nuclear γ-H2AX staining. In fact, up to 52% of cells showed XRCC1 and γ-H2AX co-staining upon 5hmC or 5fC exposure, suggesting that the majority of cells showing XRCC1 chromatin recruitment upon 5hmC or 5fC exposure, also show heightened γ-H2AX staining. To determine if other components of the BER pathway also activated during 5hmC or 5fC misincorporation, we evaluated nuclear PARylation upon treatment, and we found that both 5hmC and 5fC induced a strong nuclear PAR signal additionally to XRCC1-RFP chromatin recruitment (Figure 1D), with a large proportion of cells showing PAR^+^ XRCC1-RFP^+^ co-staining (35% and 52% for 5hmC and 5fC exposure, respectively). Coincubation of 5hmC or 5fC with the PARP inhibitor olaparib completely abolished both nuclear PARylation and XRCC1-RFP chromatin recruitment (Figure 1D), and suggest that PARP activation precedes XRCC1-RFP chromatin recruitment to promote 5hmC/5fC fixation through BER. Moreover, XRCC1-RFP positive cells largely associated with EdU staining (40% of total cells), indicating that XRCC1-RFP chromatin retention occurred mainly during S-phase (Figure 1E) and suggests that during 5hmC or 5fC misincorporation, PARP and XRCC1 are recruited to sites of 5hmC or 5fC misincorporation during S-phase. 

### 2.2. XRCC1 Maintains Genome Stability and Cell Survival upon Exposure to 5hmC

XRCC1 is a key player during BER events upon exposure to chemical agents that cause widespread oxidative damage independently of cell cycle stage. Moreover, XRCC1 associates with replication factors promote replication fork protection and restart in the absence of the replication protection factor BRCA2 [14]. To examine whether XRCC1 protected cells from DNA damage arise by misincorporation of the cytidine analogues during replication, we determine cell viability of MEFs lacking *Xrcc1*. As expected, exposure of *Xrcc1*-deficient cells to MMS, a well-known alkylating agent, markedly decreased *Xrcc1*^−/−^ cell viability compared to *wild type* cells, confirming the deficiency in BER of cells lacking *Xrcc1* (Figure 2A). We next evaluated whether XRCC1 maintained cell survival upon misincorporation of those cytidine analogues that promoted XRCC1-RFP chromatin recruitment and increased PARylation in U2OS cell system, 5hmC and 5fC. At low dose (0–5 μM), misincorporation of 5hmC decreased cell survival of *Xrcc1*^−/−^ cells compared to *wild type* (Figure 2B), whereas survival of *Xrcc1*^−/−^ cells remained unaffected upon dC or 5fC exposure. However, higher dose of 5fC also affected the survival of *Xrcc1*^−/−^ cell compared to *wild type*, suggesting that 5hmC, and 5fC to a lesser extent, challenged cell survival in the absence of *Xrcc1*. We then focused on the cytotoxic effects of 5hmC on *Xrcc1*-deficient cells. As 5hmC is a cytosine analogue naturally present in the genome, which might also be present in the nucleotide pool and incorporate during replication, we examined if 5hmC misincorporation engaged BER by examination of PARylation of nuclear proteins upon 5hmC exposure. PAR chains immunostaining of *Xrcc1*^−/−^ cells in unchallenged condition showed increased steady state levels of nuclear PARylation compared to *wild type* cells (Figure 2C) suggesting that PARP1 activity in *Xrcc1*^−/−^ cells remained active for longer or at greater extent than in *wild type* cells. Remarkably, 5hmC exposure led to a dose-dependent increased nuclear PARylation in *Xrcc1*^−/−^ but not in *wild type* cells (Figure 2C), suggesting that 5hmC misincorporation and subsequent removal promoted PARP1 activity, presumably as a consequence of SSBs arising from active 5hmC removal by BER at the replication fork. Moreover, *Xrcc1*^−/−^ cells exposed to 5hmC also showed a dose-dependent increased γ-H2AX (Figure 2C), presumably suggesting that unrepaired SSBs during 5hmC removal in the replication fork will lead to toxic DSB formation. To examine whether 5hmC misincorporation could indeed lead to DSB formation in *Xrcc1*-deficient cells, we determined the frequency of chromosomal aberrations upon 5hmC exposure. Whereas 5hmC exposure in *wild type* cells did not lead to significant changes in the frequency of chromosome aberrations, 5hmC dramatically increased the frequency of breaks, gaps and particularly those of radial structures in *Xrcc1*^−/−^ cells (Figure 2D), possibly owing to the end joining of DSBs-bearing chromosomes thus explaining the loss of *Xrcc1*^−/−^ cell viability upon exposure to 5hmC. Altogether, these data suggest that 5hmC misincorporation during DNA synthesis leads to heightened PAR and γ-H2AX, increased genomic instability and cell death in the absence of XRCC1.

### 2.3. XRCC1 Maintain Replication Fork Stability during Removal of 5hmC

It has been previously reported that XRCC1 is required to remove PARP1 from SSBs arising from MMS base lesions, thus accelerating downstream steps during BER [25]. In addition to its role during BER, XRCC1 together with DNA-PK play a crucial role at stabilising replication forks upon HU exposure [13]. Recently, XRCC1 has also been reported to promote replication fork restart by promoting extensive DNA end resection by MRE11 in the absence of BRCA2 [14]. We therefore examined whether XRCC1 was required to maintain replication fork stability upon 5hmC exposure. To this end, we performed a DNA fiber technique, whereby DNA replication was monitored by incorporation of CldU, followed by IdU in the presence of 5hmC. In *wild type* cells, 5hmC did not affect the IdU/CldU ratio (Figure 3A), suggesting that 5hmC did not perturb replication dynamics. Remarkably, 5hmC greatly reduced the IdU/CldU ratio in the absence of XRCC1 (Figure 3A), suggesting that XRCC1 is required to maintain replication fork stability upon incorporation of 5hmC. As a consequence of stalled forks, nucleases such as MRE11 or DNA2 degraded nascent DNA in a controlled manner to promote replication restart. In case of abortive repair during fork maintenance or restart, these nucleases also promote extensive nascent DNA resection leading to genomic instability. Therefore, we examined if 5hmC-mediated stalled replication forks in the absence of XRCC1 underwent extensive DNA end resection by the DNA fiber technique. Whereas *wild type* cells did not show any overt effect upon 5hmC exposure, *Xrcc1*^−/−^ cells showed a dose-dependent shortening of the IdU track (Figure 3B), suggesting that XRCC1 maintains fork integrity and restricts fork degradation upon misincorporation of 5hmC.

During BER at sites of DNA damage by MMS, PARP1 is actively recruited to SSBs to promote the recruitment of the XRCC1/LIG3α/POLβ preligation complex and boosts DNA synthesis and ligation steps in a hands-on manner [7,26]. In this context, XRCC1 was shown to be required to release PARP1 from the SSB during MMS-modified base removal. In the absence of XRCC1, PARP1 persisted on the breaks halting subsequent BER steps, whereas deletion of PARP1 completely suppressed the MMS-mediated *Xrcc1*^−/−^ cell lethality [10]. Therefore, we examine if 5hmC exposure increased PARP1 retention on chromatin during removal of 5hmC base. Cells exposed to MMS treatment greatly induced chromatin retention of PARP1 compared to untreated conditions (Figure 3C), confirming recently published results [24]. Strikingly, 5hmC also promoted PARP1 chromatin retention in both *wild type* and *Xrcc1*^−/−^ cells in a dose-dependent manner, suggesting that 5hmC misincorporation increased PARP1 chromatin association at replication forks. To address the contribution of PARP1 persistence on chromatin as a source of genomic instability in *Xrcc1*^−/−^ cells, we examined γ-H2AX levels upon low dose of 5hmC in the presence of the PARP1-trapping drug olaparib. At these given doses of 5hmC, olaparib or a combination of both in *wild type* cells did not show any increase in γ-H2AX levels. However, whereas *Xrcc1*^−/−^ cells did not show any increased γ-H2AX by single 5hmC or olaparib treatment, the combination resulted in a synergistic increased in γ-H2AX (Figure 3D), suggesting that increased trapping of PARP1 by olaparib on sites of 5hmC damage in the absence of XRCC1 induces genomic instability, measured as γ-H2AX. As XRCC1 plays dual roles at maintaining replication fork stability, and removing PARP1 from chromatin during BER, we examined whether trapped PARP1 promoted replication fork instability in the absence of XRCC1 by the DNA fiber assay. Single treatment of cells with either olaparib or low dose of 5hmC (10 μM) did not affect fork progression regardless of the genetic background, whereas combination of olaparib and 5hmC markedly reduced fork progression in the absence of XRCC1, and, to a lesser extent, *wild type* cells (Figure 3E). In addition, the heightened chromatin retention of PARP1 by 5hmC was exacerbated by olaparib treatment (Figure 3F). As a consequence of excessive PARP1chromatin retention and increased replication fork instability in the absence of XRCC1, *Xrcc1*^−/−^ cells showed a decreased viability in the presence of olaparib (Figure 3G), suggesting that upon misincorporation of 5hmC, XRCC1 prevents increased fork instability exacerbated by PARP1 trapping, probably during replication collisions with persistent PARP1 chromatin retention at BER intermediates.

### 2.4. 5hmU Recapitulates the 5hmC-Mediated Genomic Instability and Replication Fork Collapse in the Absence of XRCC1

A recent report shows that 5hmC can be taken up by cells and deaminated to 5hmU in the cytoplasm, contaminating the free nucleotide pool [27]. 5hmU is readily incorporated during DNA synthesis and removed from DNA by an active BER mechanism involving SMUG1, PARG and PARP1, suggesting that 5hmC-derived 5hmU is a novel source of genomic instability. We therefore examined if 5hmU exposure recapitulated the phenotypes observed in *Xrcc1*-deficient cells exposed to 5hmC. Consistently with the cytotoxicity by 5hmC, *Xrcc1*-deficient cells also showed decreased survival upon 5hmU exposure (Figure 4A). Similar to 5hmC, 5hmU exposure largely increased nuclear PARylation in the absence of XRCC1, at a higher level than 5hmC exposure at the same given dose (Figure 4B), probably reflecting that processing of misincorporated 5hmU in the genome induces SSBs at a larger extend than 5hmC. We also detected higher levels of γ-H2AX associated to 5hmU treatment in *Xrcc1*-deficient cells compared to *wild type*. The γ-H2AX levels achieved by 5hmU were statistically higher than those observed upon 5hmC exposure at the same dose (Figure 4B), suggesting that at a given dose, 5hmU is more cytotoxic than 5hmC. In fact, 5hmU exposure also led to heightened frequency of chromosomal aberrations (Figure 4C). Likewise, 5hmU exposure caused a moderate replication fork blockage in the absence of XRCC1 in a dose-dependent manner, whereas in *wild type* cells there was a trend without reaching statistical significance (Figure 4D), suggesting that XRCC1 also maintains replication fork stability during 5hmU misincorporation. Moreover, 5hmU exposure also led to extensive end degradation in the absence of XRCC1 (Figure 4E), and associated to heightened chromatin retention of PARP1 in a dose dependent manner, similarly to 5hmC exposure (Figure 4F). These data suggest that 5hmU inflicts replication fork defects during its misincorporation and further processing. We next examined if trapped PARP1-DNA complex by olaparib at the 5hmU DNA lesions incurred in increased genomic stability. In agreement with the role of PARP1 at signaling 5hmU DNA breaks [27], olaparib treatment largely exacerbated 5hmU-mediated γ-H2AX foci formation (Figure 5A), at a larger extent compared to the same dose of 5hmC. Consistent with the role of 5hmC at blocking ongoing replication fork progression (Figure 5B), 5hmU also hindered replication fork progression at larger extent than 5hmC exposure, which was exacerbated by olaparib treatment, thus trapping PARP1 at 5hmU sites. Finally, PARP1 trapping by increased dose of olaparib in the presence of a fixed dose of 5hmU selectively killed *Xrcc1*-deficient cells (Figure 5C), suggesting that trapped PARP1-DNA complexes in the absence of XRCC1 is detrimental for cell viability. Altogether, this work provides a novel mechanism of replication fork instability in the absence of XRCC1 by misincorporation of 5hmC and 5hmU during ongoing DNA synthesis, leading to increased PARP1 chromatin retention at BER intermediates, which leads to fork instability and cell lethality (Figure 5D).

## 3. Discussion

This work describes a cytotoxic role for 5hmC and its deamination product 5hmU upon misincorporation during replication. Whereas epigenetically modified DNA bases present in gene promoters and bodies are essential factors required for the implementation of transcriptional programmes controlling cell fate decisions and differentiation [2,28], their harmful potential to alter gene transcription upon misincorporation in the genome is not well understood. Here, we identify three epigenetically modified nucleotides as novel sources of genomic instability associated to replication fork defects and widen the scenario in which BER actively removes cytosine analogues during misincorporation by the replicative polymerases. Our results indicate that in *wild type* cells, 5hmC and its deamination product 5hmU, and 5fC to a lesser extent, activates a DNA damage response featured by increased PARylation and γ-H2AX signals, consistent with a BER mechanism activation at replication forks. Consistent with this notion, the loss of XRCC1, an essential scaffold protein functioning at MMEJ [14] at the replication fork [13] in addition to SSB repair and BER [7,26], shows a marked sensitivity to 5hmC, 5hmU and to 5fC to a lesser extent. Our data shows that the mechanism behind the aforementioned cytotoxicity is related to increased PARP1 chromatin retention at 5hmC/5hmU removal intermediates associated to replication fork collisions. In agreement with our data, a recently published work shows that contamination of the nucleotide pool by 5hmU-derived from 5hmC leads to 5hmU misincorporation in the genome, requiring some components of the BER pathway (SMUG1, PARP1, PARG) to promote cell survival [27]. The mechanism for this cytotoxicity seems to be related to olaparib trapped-PARP1 at 5hmU-processing sites persisting long enough as to be head-on collided by the replisome during the following cell cycle. However, our work uncovers that 5hmU processing at replication forks directly impact in fork dynamics, and diverges from the model proposed by Fugger et al. [27]. Our model would imply that BER processing of 5hmC or 5hmU substrates occurred straight after misincorporation, probably in the context of reversed replication forks. In such scenario, 5hmU in the nascent DNA would be processed by BER, which, in the absence of XRCC1, would lead to persistent PARP1 retention and formation of ssDNA gaps at nascent DNA, necessitating HR for replication fork restart. Several reports also describe that XRCC1 interacts with numerous bona-fide replication factors such as MRE11, PCNA, α-primase, the DNA glycosylase UNG2 or the alternative OFP pathway at replication forks [11]. Moreover, it has also been proposed that the end protection factor DNA-PKcs/KU70/KU80 regulates XRCC1 function at a subset of difficult-to-resect reversed replication forks to promote their restart [13]. We could envisage a model whereby, upon fork reversal of 5hmU-containing nascent DNA, BER would initiate at the chicken foot in nascent DNA where XRCC1 could play dual functions by coordinating PARP1 removal and preligation steps to fork resumption, regulated by PARP1 and DNA-PKcs throughout its interaction with PCNA, thus ensuring SSB fixation. Whereas PARP1 trapping seems to be the major determinant associated to fork instability during olaparib treatment, other researchers have proposed SSBs arising from BER intermediates rather than trapped PARP1 as the main cause of lethality in HR deficient cells [21]. However, it has also been reported that the marked sensitivity of *XRCC1*^−/−^ cells to the cell cycle independent alkylating agent MMS is almost completely suppressed by *PARP1*/2 deletion [25], suggesting that trapped PARP1 rather than SSBs arising from BER intermediate lesions during damaged base removal accounted for the loss of viability, implying SSBs fixation perhaps throughout postreplicative repair mechanisms. Whether these differences are due to the use of particular agents such as MMS, which engages BER across all stages of the cell cycle, or 5-hydroxymethylated nucleosides which mainly induce BER activation during replication, needs further research.

In humans, biallelic mutations in the *XRCC1* gene are associated with neurodegenerative disorders, whose symptoms encompass ocular motor apraxia, axonal neuropathy, and progressive cerebellar ataxia [5,10]. *XRCC1*-deficient patient cells show elevated PARylation and reduced efficiency of single-strand break repair. Moreover, the loss of cerebellar neurons and the ataxic phenotypes observed in *Xrcc1*^−/−^ mice were completely rescued by deletion of *Parp1*, suggesting that excessive PARP1-DNA complexes trapped at sites of endogenous DNA damage associated to BER intermediates seen in rodents may account for the neuropathological symptoms observed in these patients. Our work also uncovers that misincorporation of 5′-hydroxymethylated DNA bases at the nascent DNA is a key contributor to replication fork instability in the absence of XRCC1 and predicts that tissues containing highly enriched 5hmC-derivatives genomic DNA could be more susceptible to replication fork defects associated to collision with the BER machinery. Whether misincorporation of the epigenetically modified DNA bases 5hmC, 5fC or 5hmU during DNA repair process or during DNA synthesis in early neural progenitor cells stems from the base of these neurodegenerative disorders, paves the way for further investigations.

## 4. Materials and Methods

Transformed *wild type* and *Xrcc1* deficient murine embryonic fibroblasts (MEFs) and the pRFP-HsXRCC1 plasmid were kindly provided by Dr. Keith W. Caldecott (U. Sussex, UK). U2OS cells were transfected with HsXRCC1-RFP plasmid using TurboFect Transfection Reagent (Thermo Scientific, Waltham, MA, USA) according to the manufacturer’s protocol, and selected on G418-supplemented (1 mg/mL) growth medium. MEFs and U2OS were grown in Dulbecco’s modified Eagle’s medium (DMEM) (Gibco, Waltham, MA, USA) supplemented with 10% FBS and penicillin/streptomycin (1%, Gibco 10,000 U/mL). Cells were grown at 37 °C in a 5% CO_2_ incubator. 

2′-deoxycytidine (dC) (D3897; Sigma-Aldrich, St. Louis, MI, USA) and its derivatives 5mC (PY 7635; B&A, Dexter, MI, USA), 5hmC (PY 7588; B&A, Dexter, MI, USA), 5fC (PY 7589; B&A, Dexter, St. Louis, MI, USA), 5caC (PY 7593; B&A, Dexter, St. Louis, MI, USA), 5-chloro-2′-deoxyuridine (CldU) (Sigma-Aldrich, St. Louis, MI, USA, C6891) and 5-iodo-2′-deoxyuridine (IdU) (Sigma-Aldrich, St. Louis, MI, USA, I7125) were dissolved in PBS (10 mM stock solution) and added to cell cultures as indicated. Olaparib and 5-ethynyl-2′-deoxyuridine (EdU) were purchased from Selleckchem (Houston, TX, USA, S1060) and Baseclick (Neuried, Germany, BCN-001-100), respectively, and dissolved in DMSO (10mM stock solution). 

The antibodies used were against phospho ser139-H2AX (Millipore, Waltham, MA, USA, 05-636), PAR (Millipore, Waltham, MA, USA, MABE1016), BrdU (AbDSerotec, San Diego, CA, USA, OBT0030), BrdU (BD Bioscience, Newark, NJ, USA, 347580), Histone H3 (Invitrogen, Waltham, MA, USA, PA5-16183), PARP1 (Cell Signalling, Waltham, MA, USA, 46D11), alpha-tubulin (GeneTex, San Diego, CA, USA, GTX628802).

### 4.1. Cytotoxicity Assays

A quantity of 2.5 × 10^3^ cells were seeded in a 96-well dish and incubated at 37 °C for 3 days in the presence of the indicated 5′-modified 2′-deoxynucleoside or olaparib at the indicated doses. Cell Proliferation Kit I (MTT) (Roche, Basel, Switzerland) assay was performed according to the manufacturer’s instructions. 

### 4.2. Immunofluorescence Microscopy

For immunofluorescence analysis, 5 × 10^4^ cells were seeded in coverslips (VWR, Monroeville, PA, USA) and incubated with the chemicals for the indicated times. Cells were permeabilized with 0.25% Triton X-100 in PBS at 4 °C for 2 min and fixed with 4% formaldehyde in PBS at 4 °C for 15 min. Then, cells were blocked with 0.3% Tween20 + 3%BSA in PBS and incubated with the primary antibody of interest overnight at 4 °C. Next day, cells were incubated with alexa fluor-labelled secondary antibody and 4′6-diamidine-2-phenylindole dihydrochloride (DAPI) was added at 0.1 mg/mL for 1 h at RT in dark conditions. Finally, coverslips were mounted in ProLong^®^ Gold Antifade Reagent (Invitrogen, MA, USA). Immunofluorescence images were taken using ×40 magnification lens and quantified using the ImageJ software. Nuclear DAPI staining was used to outline cell nuclei, and a mask corresponding to nuclear DAPI area was used over the red or green channels to quantify PAR or γ-H2AX nuclear staining. Background levels were substracted from each image. For EdU immunofluorescence, cells were incubated with 10 µM EdU for 2 h. Cells were permeabilized, fixed and blocked as previously described, and click-it reaction (100 mM Tris-HCl 1.5M pH 7.5, 1 µM 488 fluorescence azide, 1 mM CuSO_4_, 100 mM ascorbic acid) was carried out for 30 min at RT in dark conditions.

### 4.3. Chromosome Aberrations Assay

To assess for chromosome aberrations, cells were grown on media containing 5hmC or 5hmU during two cell cycles. KaryoMAX colcemid solution (Gibco, MA, USA) was added at a final concentration of 0.1 μg/mL 1.5 h before collecting the cells. Cells were collected and incubated in a hypotonic solution (75 mM KCl) for 50 min at 37 °C. Fixative solution (3:1 methanol/acetic acid) was added dropwise, and cells were washed three times with this fixative solution. After washing, cells were resuspended in 500 μL of fixative solution and spread on cleaned slides dropwise. The slides were stained with Giemsa 1:10 (Sigma-Aldrich, St. Louis, MI, USA) for 30 min. To score chromosome aberrations, pictures of metaphases were scored with a DFC390 camera attached to a Leica DM6000 optical microscope using ×100 magnification lens and analyzed using LAS AF Leica software. The pictures were scored blinded for chromosome aberrations.

### 4.4. DNA Fiber Technique

For replication fork blockage purpose, 7.5 × 10^4^ cells were seeded in a 6-well dish. Cells were pulsed with 25 μM CldU for 30 min. Then, the growth medium was removed and replaced by fresh medium containing 50 μM IdU in the presence of the indicated concentration of 5hmC or 5hmU for 30 min. Cells were placed on ice, washed with cold PBS and harvested by scrapper. The cells were centrifuged at 1200 rpm 5 min 4 °C, the cell pellet resuspended in cold PBS and kept on ice. A 2 μL-drop of cells was placed on the slides, incubated for 6 min until the drop dried out. Then, 7 μL of spreading buffer was added on, mixed and incubated for 30 min at RT. DNA fibers were spread, fixed for 10 min in 3:1 methanol/acetic acid and stored at 4 °C. DNA fibers were denatured with 2.5 M HCl for 1 h, washed with PBS + 1%BSA + 0.1%Tween20, and incubated with rat-anti-BrdU antibody (1:500) for 1 h to detect CldU. The slides were washed and stained with alexa-488 anti-rat antibody for 1.5 h. Then, the slides were washed, fixed with 4% paraformaldehyde for 10 min, and stained with a mouse anti-BrdU antibody (1:500). Then, the slides were rinsed and stained with Cy3 anti-mouse antibody for 2 h. Finally, the slides were mounted in Fluoromount-G (eBioscience, San Diego, CA, USA). Pictures of fibers were scored using ×100 magnification lens.

### 4.5. Chromatin Retention Assay and Immunoblotting

PARP1 chromatin retention assay was performed essentially as described by Demin and colleagues [25].

### 4.6. Statistical Analysis

The number of independent technical repeats (n) are indicated in figure legends. Data are shown as mean ± standard error of the mean (s.e.m.) or median. The nonparametric Mann–Whitney test or Student’s *t*-test were employed to determine statistical significance (* *p* < 0.05, ** *p* < 0.01, *** *p* < 0.001, **** *p* < 0.0001). Data analyses were performed in GraphPad Prism.

## Figures and Tables

**Figure 1 ijms-23-00893-f001:**
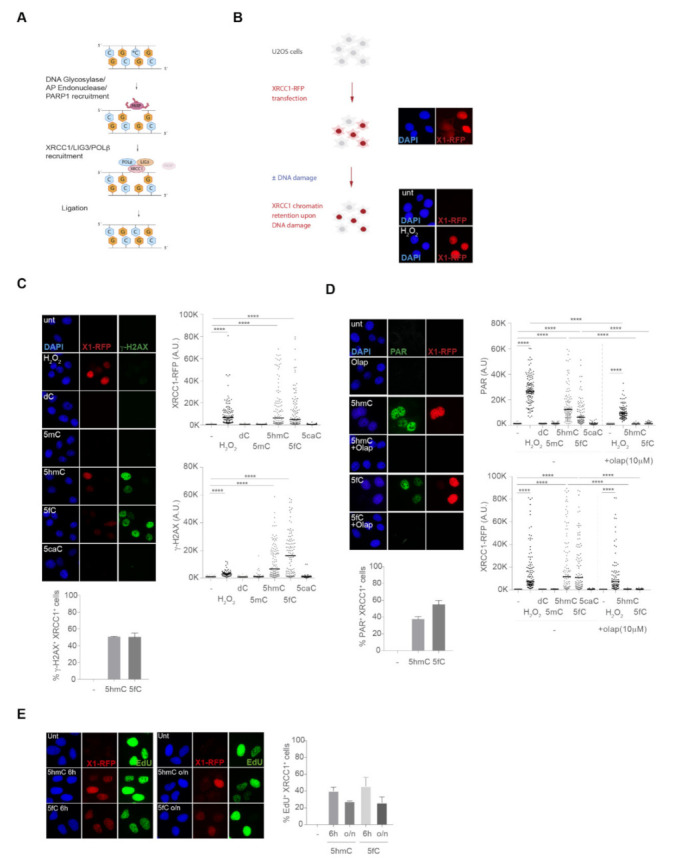
The epigenetically modified DNA bases 5hmC and 5fC activates the BER pathway. (**A**) Representative scheme of BER-SSBR pathway. An epigenetically modified cytosine is recognized by a DNA glycosylase, which creates an abasic site. AP endonuclease 1 (APE1) recognizes and cleaves the sugar-phosphate backbone of the abasic site, generating a single strand break (SSB), which is detected by PARP1. PARP1 acts like a sensor of SSBs and generates poly(ADP-ribose) chains, which interact with the BRCT domain of XRCC1 to promote its recruitment. XRCC1 displaces PARP1 and acts as a scaffold protein for the recruitment of POLβ, LIGIII-α, and several end-processing factors (not depicted) to seal the nick present in the broken strand. (**B**) Left, scheme depicting the generation of HsXRCC1-RFP overexpressing U2OS cells and its chromatin retention assay by DNA damage after incubation with H_2_O_2_. Right, representative immunofluorescence images of fixed and permeabilized, or permeabilized and fixed HsXRCC1-RFP chromatin retention assay upon incubation with H_2_O_2_ (10 mM). (**C**) Left, chromatin retention assay of HsXRCC1-RFP fusion protein and γ-H2AX after exposure to cytosine analogues (240 µM) for 16 h in HsXRCC1-overexpressing U2OS cells. H_2_O_2_ (10 mM) was added for 10′ followed by 15′ recovery in fresh media before permeabilization. Right, quantification of HsXRCC1-RFP levels per nucleus (top right), and γ-H2AX intensity signal per nucleus (bottom right) (*n* = 100 of each of 3 biological replicates, Mann–Whitney test; line represents median value). Bottom, percentage of γ-H2AX^+^ XRCC1^+^ co-stained cells by 5hmC or 5fC exposure. (**D**) Left, immunofluorescence showing the chromatin retention of HsXRCC1-RFP and poly(ADP-ribose) (PAR) after exposure to 5hmC or 5fC (240 µM) for 16 h, in the presence or absence of the PARP inhibitor olaparib (10 µM), in HsXRCC1-overexpressing U2OS cells. Right, plot depicting PAR intensity signal per nucleus (top right), and HsXRCC1-RFP levels per nucleus (bottom right) (*n* = 90 of each of 3 biological replicates, Mann–Whitney test; line represents median value). Bottom, percentage of PAR^+^ XRCC1^+^ co-stained cells by 5hmC or 5fC exposure. (**E**) Left, immunofluorescence showing the chromatin retention of HsXRCC1-RFP and EdU co-staining after exposure to 5hmC or 5fC (240 µM) for 6 h or 16 h, in HsXRCC1-overexpressing U2OS cells. Right, bar plot depicting the percentage of EdU^+^ HsXRCC1-RFP^+^ cells (*n* = 150 of each of 3 biological replicates, mean ± s.e.m.).

**Figure 2 ijms-23-00893-f002:**
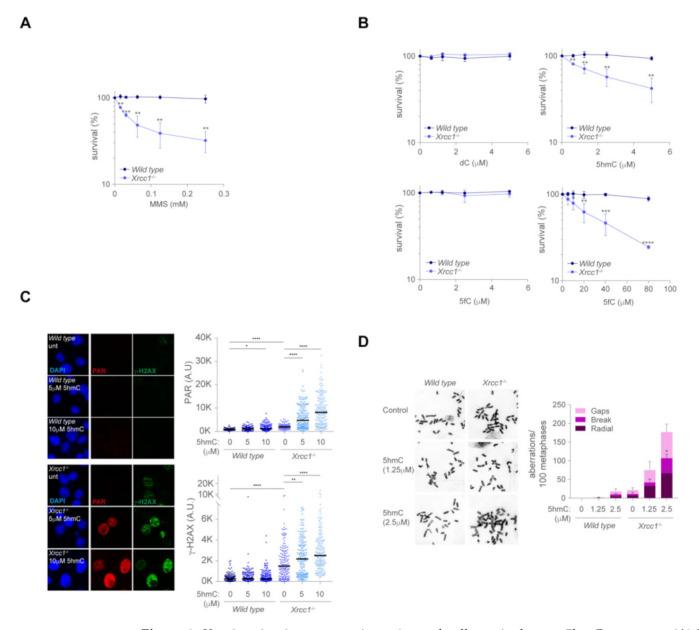
*Xrcc1* maintains genome integrity and cell survival upon 5hmC exposure. (**A**) MTT survival assay of *wild type* and *Xrcc1*^−/−^ MEFs treated with methyl methanosulfonate (MMS) at indicated doses for 3 days (*n* = 3, mean ± s.e.m). (**B**) MTT survival assay of *wild type* and *Xrcc1*^−/−^ MEFs treated with 5C, 5hmC and 5fC at indicated doses for 3 days (*n* = 3, mean ± s.e.m). (**C**) Left, representative immunofluorescence images showing PAR nuclear staining (red) and γ-H2AX (green) of *wild type* and *Xrcc1*^−/−^ MEFs upon exposure to 5hmC (5 µM and 10 µM) for 3 h. DAPI (blue) stains nuclear DNA. Right, plot depicting PAR intensity signal per nucleus (top right) and γ-H2AX intensity signal per nucleus (bottom right) (*n* = 150 of each of 3 biological replicates, Mann–Whitney test; central line represents median value). (**D**) Left, representative images of chromosome aberrations from *wild type* and *Xrcc1*^−/−^ MEFs following 5hmC treatment (1.25 µM and 2.5 µM) for two cell cycles. Right, a bar plot of breakdown of the different types of chromosomal aberrations (*n* = 50 of each of 3 biological replicates, Student’s *t*-test; bar represents mean ± s.e.m.).

**Figure 3 ijms-23-00893-f003:**
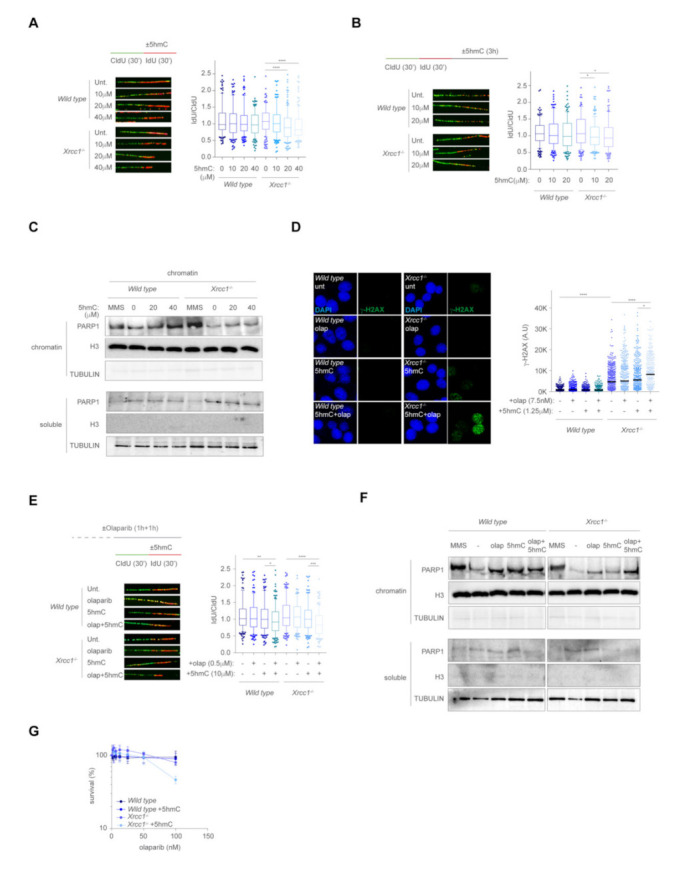
5hmC induces replication fork instability, exacerbated upon PARP1 trapping by olaparib. (**A**) Top left, scheme of the DNA fiber assay. Bottom left, representative images of DNA fibers from *wild type* and *Xrcc1*^−/−^ MEFs after increasing doses of 5hmC (10 µM, 20 µM and 40 µM) exposure. Right, box plot of the frequency of IdU/CldU ratios (*n* = 150 of each of 2 biological replicates, Mann–Whitney test; box central line represents median and boxes include 10–90 percentile of data points). (**B**) Top left, scheme of DNA resection by the fiber assay. Bottom left, representative images of DNA fibers from *wild type* and *Xrcc1*^−/−^ MEFs after increasing doses of 5hmC (10 µM, and 20 µM) for 3 h upon IdU incorporation. Right, box plot of the frequency of IdU/CldU ratios (*n* = 150 of each of 2 biological replicates, Mann–Whitney test; box central line represents median and boxes include 10–90 percentile of data points). (**C**) Western blot of PARP1, histone H3 and tubulin levels of chromatin and soluble-containing fractions from *wild type* and *Xrcc1*^−/−^ MEFs upon exposure to increasing doses of 5hmC (20 µM and 40 µM) for 3 h. MMS was used as a control treatment. (**D**) Top, representative immunofluorescence images showing γ-H2AX nuclear staining in *wild type* and *Xrcc1*^−/−^ MEFs upon exposure to 5hmC (1.25 µM), olaparib (7.5 nM) or 5hmC+olaparib for 3 h. Bottom, dot plot showing the quantitation of nuclear γ-H2AX intensity signal in *wild type* and *Xrcc1^−/−^* MEFs upon treatment with 5hmC (1.25 µM), olaparib (7.5 nM) or 5hmC+olaparib for 3 h (*n* = 150 of each of 3 biological replicates, Mann–Whitney test; central line represents median value). (**E**) Top left, scheme of the DNA fiber assay of *wild type* and *Xrcc1*^−/−^ MEFs in the presence of 5hmC alone or in combination with olaparib. Bottom left, representative images of DNA fibers from *wild type* and *Xrcc1*^−/−^ MEFs after 5hmC (10 µM) exposure alone or in combination with olaparib (0.5 µM). Right, box plot of the frequency of IdU/CldU ratios (*n* = 150 of each of 3 biological replicates, Mann–Whitney test; box central line represents median and boxes include 10–90 percentile of data points). (**F**) Western blot of PARP1, histone H3 and tubulin levels in of chromatin and soluble-containing fractions of *wild type* and *Xrcc1*^−/−^ MEFs upon exposure to 5hmC (20 µM), olaparib (0.5 µM) and 5hmC plus olaparib for 3 h. MMS was used as a control treatment. (**G**) MTT survival assay of *wild type* and *Xrcc1*^−/−^ MEFs exposed to the indicated doses of olaparib alone or combined with a fixed dose of 5hmC (0.3 µM) for 3 days (*n* = 3, mean ± s.e.m).

**Figure 4 ijms-23-00893-f004:**
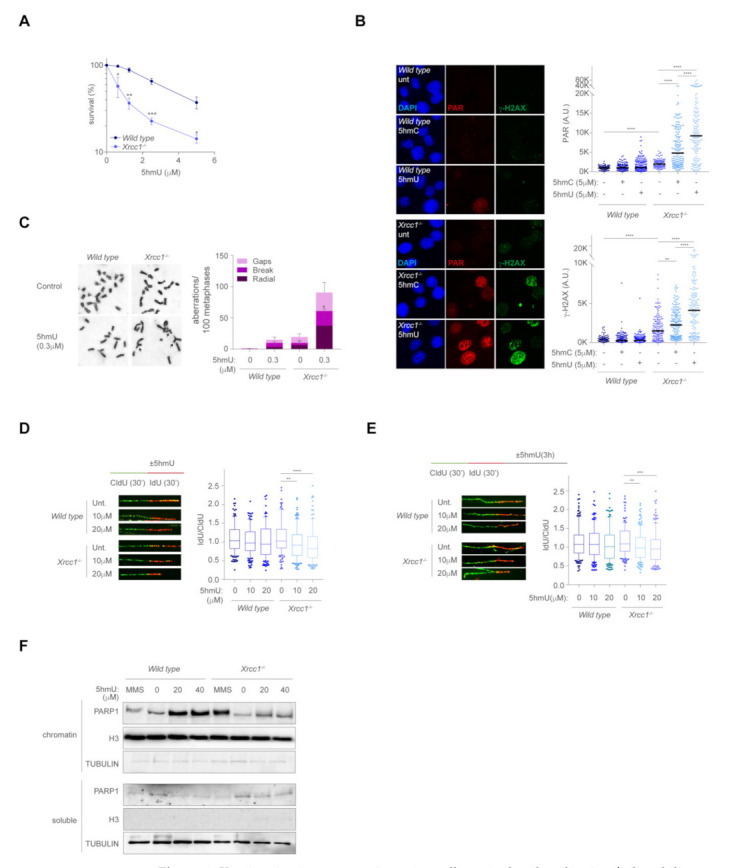
*Xrcc1* maintains genome integrity, cell survival and replication fork stability upon 5hmU exposure. (**A**) MTT survival assay of *wild type* and *Xrcc1*^−/−^ MEFs treated with 5hmU at indicated doses for 3 days (*n* = 3, mean ± s.e.m). (**B**) Left, representative immunofluorescence images showing PAR nuclear staining (red) and γ-H2AX (green) of *wild type* and *Xrcc1*^−/−^ MEFs upon exposure to 5hmC and 5hmU (5 µM) for 3 h. DAPI (blue) stains nuclear DNA. Right, plot depicting PAR intensity signal per nucleus (top right) and γ-H2AX intensity signal per nucleus (bottom right) (*n* = 150 of each of 3 biological replicates, Mann–Whitney test; central line represents median value). (**C**) Left, representative images of chromosome aberrations from *wild type* and *Xrcc1^−/−^* MEFs following 5hmU treatment (0.3 µM) for two cell cycles. Right, a bar plot of breakdown of the different types of chromosomal aberrations (*n* = 50 of each of 2 biological replicates, Student’s t-test; bar represents mean ± s.e.m.). (**D**) Top left, scheme of the DNA fiber assay. Bottom left, representative images of DNA fibers from *wild type* and *Xrcc1*^−/−^ MEFs after increasing doses of 5hmU (10 µM and 20 µM) exposure. Right, box plot of the frequency of IdU/CldU ratios (*n* = 150 of each of 3 biological replicates, Mann–Whitney test; box central line represents median and boxes include 10–90 percentile of data points). (**E**) Top left, scheme of DNA resection by the fiber assay. Bottom left, representative images of DNA fibers from *wild type* and *Xrcc1*^-/-^ MEFs after increasing doses of 5hmU (10 µM, and 20 µM) for 3 h upon IdU incorporation. Right, box plot of the frequency of IdU:CldU ratios (*n =* 150 of each of 2 biological replicates, Mann-Whitney test; box central line represents median and boxes include 10-90 percentile of data points). (**F**) Western blot of PARP1, histone H3 and tubulin levels of chromatin and soluble-containing fractions from *wild type* and *Xrcc1^−/−^* MEFs upon exposure to increasing doses of 5hmU (20 µM and 40 µM) for 3 h. MMS was used as a control treatment.

**Figure 5 ijms-23-00893-f005:**
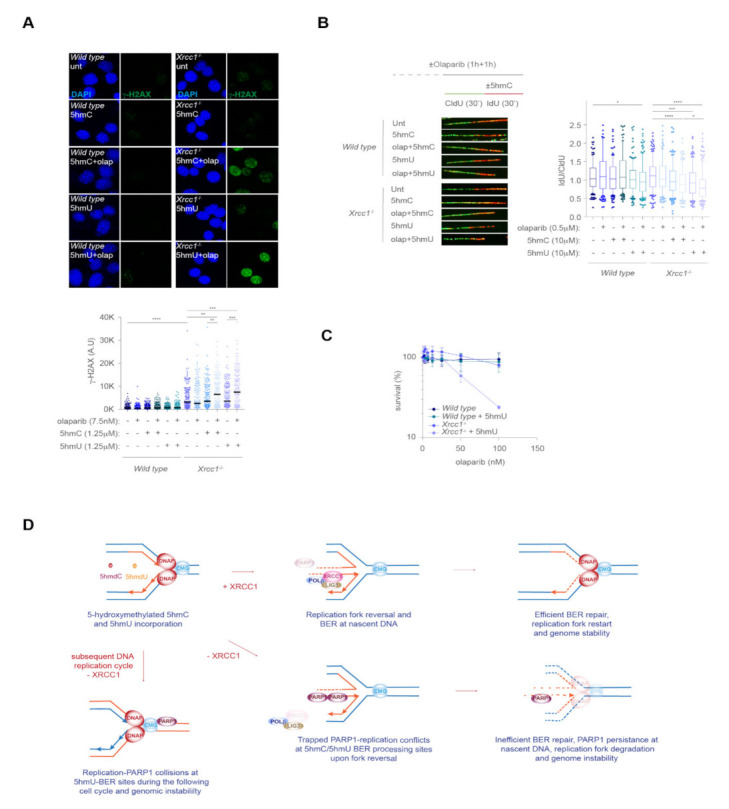
Trapped *Parp1* by olaparib exacerbates 5hmU genomic instability in the absence of *Xrcc1.* (**A**) Top, representative immunofluorescence images showing γ-H2AX nuclear staining in *wild type* and *Xrcc1*^−/−^ MEFs after 5hmC or 5hmU (1.25 µM) exposure alone or in combination with olaparib (7.5 nM) for 3 h. Bottom, dot plot showing the quantitation of nuclear γ-H2AX intensity signal in *wild type* and *Xrcc1*^−/−^ MEFs upon treatment with 5hmC or 5hmU (1.25 µM), olaparib (7.5 nM) or 5hmC/5hmU plus olaparib for 3 h (*n* = 150 of each of 3 biological replicates, Mann–Whitney test; central line represents median value). (**B**) Top left, scheme of the DNA fiber assay of *wild type* and *Xrcc1*^−/−^ MEFs in the presence of 5hmC or 5hmU in combination with olaparib. Bottom left, representative images of DNA fibers from *wild type* and *Xrcc1*^−/−^ MEFs after 5hmC or 5hmU (10 µM) exposure alone or in combination with olaparib (0.5 µM). Right, box plot of the frequency of IdU/CldU ratios (*n* = 150 of each of 3 biological replicates, Mann–Whitney test; box central line represents median and boxes include 10–90 percentile of data points). (**C**) MTT survival assay of *wild type* and *Xrcc1*^−/−^ MEFs exposed to the indicated doses of olaparib alone or combined with a fixed dose of 5hmU (0.3 µM) for 3 days (*n* = 3, mean ± s.e.m). (**D**) Scheme depicting the proposed model for XRCC1 function during replication fork collisions with trapped Parp1 at BER intermediate DNA structures during removal of 5-hydroxymethylated DNA bases. Upon incorporation of 5hmC or 5hmU, these modified DNA bases are removed by BER glycosylases followed by APE1 incision of the sugar-phosphate backbone, leaving a SSB potentially harmful for replication fork progression. Upon binding to the SSB, XRCC1 is recruited by PARP1 to accelerate the fixation of the damaged base thus promoting replication fork resumption. In the absence of XRCC1, persisting PARP1 at SSBs generated during removal of 5hmC or 5hmU by BER, hinders replication fork progression leading to genomic instability.

## Data Availability

This study does not include any genome-wide NGS, RNA-seq or data to be deposited in any repository. Any piece of data developed in this MS, as well as cell lines or reagents, are fully available upon request by any researcher.

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
