# Peer review of "XRCC1 Prevents Replication Fork Instability during Misincorporation of the DNA Demethylation Bases 5-Hydroxymethyl-2′-Deoxycytidine and 5-Hydroxymethyl-2′-Deoxyuridine"

_ijms, 2022, doi:10.3390/ijms23020893_

Round 1

Reviewer 1 Report

Comments to the authors:

The authors present in their manuscript interesting findings on the Base Excision Repair mechanism and propose a replication fork instability mechanism in the absence of Xrcc1, induced by misincorporation during DNA synthesis, upon treatment with 5-hydroxymethyl-2'-deoxycytidine (5hmC) and/or 5-hydroxymethyl-2'-deoxyuridine (5hmU). Their suggestion is supported by results showing increased poly[ADP-ribose] polymerase (Parp)1 chromatin retention at the produced BER intermediates. However, in the view of the assessments found and provided below, I really hesitate to recommend this paper for International Journal of Molecular Sciences yet.

General comments:

- It should be noted that the Parp trapping ability of 5hmU has already been demonstrated as mentioned also in the manuscript. 5hmU in cells is formed under oxidative stress conditions (see reference 24) or as deamination product from 5hmC (Cell. 2011 145, 423), thus both lesions are related and an expert in the field would expect similar behavior, as it was found by the authors (see also reference 26). For this reason, the authors should demonstrate in a clear manner how their results differentiate from those found in the related papers that recently appeared in the literature.

- The authors should provide an explanation on their strategy on selection of the specific 2’-deoxycytidine derivatives/lesions for their studies.

- There is a confusion in the manuscript regarding the 2’-deoxycytidine, cytosine, cytidine and the corresponding modified 2’-deoxynucleodes that were used in these studies. Moreover, the chemical structures of all nucleosides utilized, even the names of some of them [see 5fC (PY 7589; B&A) and 5caC (PY 7593; B&A)], are missing. For some reason 2’-deoxycytidine is given as 5C while the commonly used abbreviation is dC or 2’-dC.

Comments on the manuscript:

- The references numbering in the manuscript starts from ref. 2 and is given before the fullstop. Ref 1 is given in the abstract.

- A XRCC1 should be given in a consistent manner as XRCC1 or Xrcc1 or Xrcc1 throughout the text.

- “oC” is underlined and needs to be corrected to “oC” throughout the text

- p.1, Line 9 “…. cytosine demethylation process” ….reference is missing.

- p.2, Line 3 from the end “CO2 incubator” 2, font needs to be changed (subscript)

- p.2, Line 1-2 from the end…..the names and the chemical structures are missing for 5C (D3897; Sigma), 5mC (PY 7635; B&A), 5hmC (PY 7588; B&A), 5fC (PY 7589; B&A); 5caC (PY 7593; B&A), CldU (Sigma, C6891), IdU (Sigma, I7125). 5C abbreviation needs to be corrected.

- it is frequently found the phase “of the drug”, “the drug”. A precise description is missing.

p.3 in “2.3 Chromosome aberrations assay”, line 2, “One and a half hour before” should be changed to “1.5h”. Time should be given in a consistent manner throughout the text.

p.3 “2.4. DNA fiber technique” line 7 “mix”….. typo should be corrected to “mixed”

p.4 “3. Results”, line 4…. should be rephrased to avoid using the word “upon” twice

p.5 line 5 “cytosine or 5fC exposure”….was cytosine used instead of dC?

p.5 line 3 from the end “we examine if 5hmC cause” should be changed to “we examined if 5hmC caused”

p.6 line 1 “published results”…. reference is missing.

p.6 line 22 “could be at least in part be explained by” needs rephrasing

p.6 line 27 “can be taken by cells” should be corrected to “can be taken up by cells”

p.6 “to Xrcc1 deficiency could be at least in part be explained by the increased fork instability” needs to be rephrased

In most figures the concentrations of H2O2 or the nucleosides due to typos or errors during conversion to pdf, are given in M instead of μM or nM, etc.

Author Response

Rebuttal letter

Reviewer 1:

The authors present in their manuscript interesting findings on the Base Excision Repair mechanism and propose a replication fork instability mechanism in the absence of Xrcc1, induced by misincorporation during DNA synthesis, upon treatment with 5-hydroxymethyl-2'-deoxycytidine (5hmC) and/or 5-hydroxymethyl-2'-deoxyuridine (5hmU). Their suggestion is supported by results showing increased poly[ADP-ribose] polymerase (Parp)1 chromatin retention at the produced BER intermediates. However, in the view of the assessments found and provided below, I really hesitate to recommend this paper for International Journal of Molecular Sciences yet.

We acknowledge reviewer 1 for the critical reading of the MS, and for the concern described here, which will help us to undoubtedly improve our MS. We have now addressed most of his/her concerns, and have included a most comprehensive discussion about how our data differ from Fugger´s one, hoping that the new version of the MS will satisfy reviewer 1´s criticisms.

General comments:

- It should be noted that the Parp trapping ability of 5hmU has already been demonstrated as mentioned also in the manuscript. 5hmU in cells is formed under oxidative stress conditions (see reference 24) or as deamination product from 5hmC (Cell. 2011 145, 423), thus both lesions are related and an expert in the field would expect similar behavior, as it was found by the authors (see also reference 26). For this reason, the authors should demonstrate in a clear manner how their results differentiate from those found in the related papers that recently appeared in the literature.

We thank Reviewer 1 for this helpful comment. Our MS describes the role of XRCC1 during the removal of 5hmdC or 5hmdU from nascent DNA, whereas the work by Guo et al. (Cell. 2011, 145) focused on the role of BER during removal of 5hmC from the template DNA strand, our work uncovers a role for XRCC1 during removal of misincorporated 5hmdC or 5hmdU on the nascent DNA strand. Our research MS also diverts from the one described by Fugger et al. in relationship with the mechanistic contribution of Parp1-trapped DNA lesions to genomic instability. Whereas Fugger et al. proposed that PARP1-trapped at 5hmU-processing BER intermediates ocurring during S phase are potential impedements for the replisome during the following cell cycle, our data suggest that short exposure (30´) and misincorporation of 5hmdC/5hmdU induces replication fork stalling/blockage in Xrcc1-/- cells during fork progression. To us, these data suggest that 5hmdC/5hmdU readily misincorporates in the nascent DNA strand and undergo post-replicative fixation by BER, which are encountered by the replisome. How could this happen? Our current model of work, is that BER intermediates involving trapped PARP1 postreplicatively can collide with reversed DNA polymerases (i.e: reversed polymerase upon TRCs). In this situation, DNA polymerases would encounter BER intermediates during reversal which would require XRCC1 for PARP1 removal to avoid fork disassembly and degradation. We are currently studying the contribution of the reversal ATPases to this process by knocking them down in the WT and XRCC1 cell lines, in a situation similar to the one described by Mijic et al. 2017. during olaparib treatment in siBRCA2 cells, whereby chromosome instability and extensive end resection in Brca2 KD cells was promoted by fork reversal.

- The authors should provide an explanation on their strategy on selection of the specific 2’-deoxycytidine derivatives/lesions for their studies.

We thank R1 for this comment. Our strategy on selection of the specific 2´-deoxycytidine derivatives originated from the cytotoxic effects of formaldehyde and its DNMT-mediated cytidine derivative 5hmC, a well-known cytosine demethylation intermediate. We have data suggesting that Fanconi deficient cells are hypersensitive to 5hmC (and to 5hmU) but not to any other cytidine analogues (currently under revision in Cell Death and Disease). Due to the main role of BER during cytosine demethylation by removing 5fC and 5caC, we reasoned that BER and Fanconi Anemia pathway could play independent roles during cytosine demethylation. That was the main reasoning for our selection.

- There is a confusion in the manuscript regarding the 2’-deoxycytidine, cytosine, cytidine and the corresponding modified 2’-deoxynucleodes that were used in these studies. Moreover, the chemical structures of all nucleosides utilized, even the names of some of them [see 5fC (PY 7589; B&A) and 5caC (PY 7593; B&A)], are missing. For some reason 2’-deoxycytidine is given as 5C while the commonly used abbreviation is dC or 2’-dC.

We thank R1 for his/her comment. We have now gone through the text to standardize the nucleoside nomenclature. We have also included information of the referenced products.

Comments on the manuscript:

- The references numbering in the manuscript starts from ref. 2 and is given before the fullstop. Ref 1 is given in the abstract.

We thank R1 for this comment. Ref1 has now been included in the main text and deleted from the abstract.

- A XRCC1 should be given in a consistent manner as XRCC1 or Xrcc1 or Xrcc1 throughout the text.

We acknowledge R1 for his/her comment. In accordance to HGNC guidelines, we have now corrected this issue by giving “XRCC1” or “Xrcc1” for the human or mouse gene respectively, and “XRCC1” for the protein independently of its origin.

- “oC” is underlined and needs to be corrected to “oC” throughout the text

This has been corrected through the main text

- p.1, Line 9 “…. cytosine demethylation process” ….reference is missing.

Reference has been added. (Manning, Kohli. ACS Chem Biol 2012)

- p.2, Line 3 from the end “CO2 incubator” 2, font needs to be changed (subscript)

Thank you for spotting this. It has now been subscripted.

- p.2, Line 1-2 from the end…..the names and the chemical structures are missing for 5C (D3897; Sigma), 5mC (PY 7635; B&A), 5hmC (PY 7588; B&A), 5fC (PY 7589; B&A); 5caC (PY 7593; B&A), CldU (Sigma, C6891), IdU (Sigma, I7125). 5C abbreviation needs to be corrected.

I apologize for overlooking annotating these reagents. This issue has now been amended. 5C has been replaced by dC throughout the text.

- it is frequently found the phase “of the drug”, “the drug”. A precise description is missing. This have now been corrected.

A precise description of “the drug” has been described accordingly.

p.3 in “2.3 Chromosome aberrations assay”, line 2, “One and a half hour before” should be changed to “1.5h”. Time should be given in a consistent manner throughout the text.

“Hours” has been changed consistently

p.3 “2.4. DNA fiber technique” line 7 “mix”….. typo should be corrected to “mixed”

Done

p.4 “3. Results”, line 4…. should be rephrased to avoid using the word “upon” twice

Thank you for letting us know this redundancy, it has now been changed

p.5 line 5 “cytosine or 5fC exposure”….was cytosine used instead of dC?

I apologize for the wording used, which could sometimes be misleading for an expert in this field. We wrote our manuscript to target a general or cell and molecular biology field audience rather than for a chemistry audience, and it was probably misleading. All cytidine derivatives used in this study are 2´-deoxynucleosides modified at 5´. 5hmC and 5hmU abbreviations stand for “5-hydroxymethyl-2’-deoxycytidine” and “5-hydroxymethyl-2’-deoxyuridine” respectively, but abbreviated as 5hmC or 5hmU to ease the understanding of the MS by the readership.

p.5 line 3 from the end “we examine if 5hmC cause” should be changed to “we examined if 5hmC caused”

We thank R1 for this annotation. This now has been changed in the text.

p.6 line 1 “published results”…. reference is missing.

Ref26 has now been added to it.

p.6 line 22 “could be at least in part be explained by” needs rephrasing

This sentence has now been toned down and rephrased

p.6 line 27 “can be taken by cells” should be corrected to “can be taken up by cells”

Thank you for letting us know this mistake. We have now added “up” to the

p.6 “to Xrcc1 deficiency could be at least in part be explained by the increased fork instability” needs to be rephrased

This sentence has now been toned down and rephrased

In most figures the concentrations of H2O2 or the nucleosides due to typos or errors during conversion to pdf, are given in M instead of μM or nM, etc.

I have to apologize for the typos and for the poor quality of graphs throughout the MS. The MS we upload contained a high-resolution images, and symbols were properly uploaded. However, during the conversion of the MS to the reviewers´ version, a formatting issue must have happened. We have now uploaded a high-res MS and figures, and typos have been corrected by the right symbols. We hope the new version of the MS facilitates its reading and comprehension.

Reviewer 2 Report

In this manuscript, Pena-Gomez and colleagues analyse the function of XRCC1 regarding replication fork stability when two demethylated cytidine analogues, 5hmC and 5hmU, are misincorporated into nascent strands. Their results bring interesting new insight on the role of XRCC1, known to be involved in the Base Excision Repair (BER) pathway and replisome stability, on the replication-associated processing of naturally occuring bases modifications that are therefore potential endogenous threats for genetic stability.

It has to be noticed that Fugger and colleagues (reference 26 from this manuscript) already demonstrated the toxicity of 5hmU incorporated in the genome when processed by BER (through SMUG1) in cells treated with PARP inhibitors, leading to PARP1 trapping and replication fork collapse, which represent the main findings of the present study. However, the novelty here is that such effect arise during nucleotide misincorporation by the replicative polymerase, rather than when the replication fork encounters a pre-existing DNA lesion with trapped PARP. Despite these data represent novelty and are overall interesting and suitable for publication in IJMS, they are not presented in this way. Besides, few other points must also be improved.

Major points:

1) From a mechanistic point of view, I have problems with authors interpretations. They claim that 5hmC and 5hmU are incorporated in the newly syntethized strand during replication before to be processed by BER machinery involving PARP1 and XRCC1 functions, and that XRCC1 deficiency induces PARP1 trapping to BER intermediates, leaving an unrepaired SSB that eventually leads to DSB formation and fork instability. However, they  conclude that "the phenotypes associated to Xrcc1 deficiency could be at least in part be explained by the increased fork instability owing to replication collisions with persistent Parp1 chromatin retention at BER intermediates" (page 6, last sentence before 3.4). If 5hmC/5hmU are  incorporated, how could a replication fork collide with an already replicated locus? This point is essential for all the subsequent conclusions. While Fugger and colleagues exposed cells to 5hmU for at least one cell cycle before to observe fork instability, allowing 5hmU to be present in the chromatin before passage of the replication fork, the present study demonstrate similar fork instability upon 5hmU incorporation. This raises interesting questions regarding the role of XRCC1 at the replication fork, and how XRCC1 physical interactions with replications factors could regulate such processes. This also challenges the paradigm of lesion bypass mechanisms that prevent replication fork instability. These points should be at least discussed.

2) Information is missing in the methods section:

  • Generation and culture of RFP-tagged XRCC1 U2OS cells
  • Immunofluorescence protocol is not described from the blocking, and more importantly, how the fluorescence signals are quantified is not explained.

3) Last sentences of the different paragraphs of the result section represent interpretations that are not totally supported by the data in the sense that more direct evidences would be necessary, and might therefore be partly integrated in the discussion section.

4) Figure 1C-D: it could be interesting to show the proportion of positive cells including co-staining. Moreover, use of an S-phase marker or a sensor of arrested replication forks (like FANCD2) would be relevant to support the conclusions.

5) Chromatin retention assays and immunoblotting Figures 3 and 4 : the soluble fraction has to be included. The amount of PARP1 at chromatin cannot be appreciated with H3 blotting but require to show unbound PARP1.

6) Figure 5A and B: Authors conclude that 5hmU induces more gH2AX and replication fork arrest than 5hmC (see result section). However, this is not clear regarding the dot plots, and there are no statistical analyzes in this way to support these conclusions.

7) Lot of information is missing in the figure legends, especially for drugs concentrations with empty spaces before "M". What is 5C Fig 2B (graph and figure legend)? Nothing about Fig. 4D? Authors should carrefully proofread this part.

Minor points:

  • Some acronyms must be explained: TET, 5fC, 5caC, TDG
  • Methods: phospho ser139-H2AX
  • Fig 3C: the difference of staining between 5hmC+olap and the other conditions in XRCC1-/- cells is huge in the picture compared to quantifications in the dot plot.
  • Discussion: it is written that "In agreement with our data, a recently published work shows that contamination of the nucleotide pool by 5hmU-derived from 5hmC leads to 5hmU mis- incorporation in the genome, requiring some components of the BER pathway (SMUG1, PARP1, PARG) to promote cell survival". However, this work (Fugger 2021) focus on 5hmU already present in the chromatin and not newly incorporated.

Author Response

Reviewer 2:

In this manuscript, Pena-Gomez and colleagues analyse the function of XRCC1 regarding replication fork stability when two demethylated cytidine analogues, 5hmC and 5hmU, are misincorporated into nascent strands. Their results bring interesting new insight on the role of XRCC1, known to be involved in the Base Excision Repair (BER) pathway and replisome stability, on the replication-associated processing of naturally occuring bases modifications that are therefore potential endogenous threats for genetic stability.

It has to be noticed that Fugger and colleagues (reference 26 from this manuscript) already demonstrated the toxicity of 5hmU incorporated in the genome when processed by BER (through SMUG1) in cells treated with PARP inhibitors, leading to PARP1 trapping and replication fork collapse, which represent the main findings of the present study. However, the novelty here is that such effect arise during nucleotide misincorporation by the replicative polymerase, rather than when the replication fork encounters a pre-existing DNA lesion with trapped PARP. Despite these data represent novelty and are overall interesting and suitable for publication in IJMS, they are not presented in this way. Besides, few other points must also be improved.

We thank R2 for his/her helpful comment and for the critical reading of the MS. As pointed out by R2, the findings here relate to the effects of 5hmC/5hmU misincorporation and further processing by SSBR/BER during ongoing replication fork causing fork instability, rather than head-on collisions of the replisome with pre-existing PARP1-DNA lesions during the following cell cycle. We believe that these findings divert from the one published by Fugger et al. as to deserve publication at IJMS. Whereas Fugger et al. proposed that PARP1-trapped at 5hmU-processing BER intermediates occurring during S phase are potential impediments for the replisome during the following cell cycle, our data suggest that short exposure (30´) and misincorporation of 5hmdC/5hmdU induces replication fork stalling/blockage in Xrcc1-/- cells during fork progression. To us, this data suggests that 5hmdC/5hmdU readily misincorporates in the nascent DNA strand and undergo post-replicative fixation by BER, which are encountered by the replisome. How could this happen? Our current model of work, is that post-replicatively occurring BER intermediates involving trapped-PARP1 can collide with reversed DNA polymerases (i.e: reversed polymerase upon TRCs). In this situation, DNA polymerases would encounter BER intermediates during reversal which would require XRCC1 for PARP1 removal to avoid fork disassembly and degradation. We are currently studying the contribution of the reversal ATPases to this process by knocking them down in the WT and XRCC1 cell lines, in a situation similar to the one described by Mijic et al. 2017. during olaparib treatment in siBRCA2 cells, whereby chromosome instability and extensive end resection in Brca2 KD cells was promoted by fork reversal.

An alternative model would be that XRCC1 not only would serve as a scaffold protein for the BER preligation complex but also would be a sensor to couple completion of postreplicative BER during removal of 5hmU to replication fork progression. In the presence of XRCC1, PARP1 would be remove from chromatin and ligation would procced. In its absence, MRE11 would be recruited and extensive end resection would occur, to promote restart together with DNAPK as suggested by Ying et al 2016. However to date, we have no data on this hypothesis. We have now included this in the discussion.

Major points:

1) From a mechanistic point of view, I have problems with authors interpretations. They claim that 5hmC and 5hmU are incorporated in the newly syntethized strand during replication before to be processed by BER machinery involving PARP1 and XRCC1 functions, and that XRCC1 deficiency induces PARP1 trapping to BER intermediates, leaving an unrepaired SSB that eventually leads to DSB formation and fork instability. However, they conclude that "the phenotypes associated to Xrcc1 deficiency could be at least in part be explained by the increased fork instability owing to replication collisions with persistent Parp1 chromatin retention at BER intermediates" (page 6, last sentence before 3.4). If 5hmC/5hmU are incorporated, how could a replication fork collide with an already replicated locus? This point is essential for all the subsequent conclusions. While Fugger and colleagues exposed cells to 5hmU for at least one cell cycle before to observe fork instability, allowing 5hmU to be present in the chromatin before passage of the replication fork, the present study demonstrate similar fork instability upon 5hmU incorporation. This raises interesting questions regarding the role of XRCC1 at the replication fork, and how XRCC1 physical interactions with replications factors could regulate such processes. This also challenges the paradigm of lesion bypass mechanisms that prevent replication fork instability. These points should be at least discussed.

We entirely agree with Reviewer 2, and he/her clearly spotted the differences in the molecular mechanisms of replication fork instability by misincorporation of 5hmC/5hmU in Xrcc1-/- cells described in our MS, with the one previously reported (Fugger et al. Science 2021). One possibility could be related to persistent PARP1 at SSB arising from 5hmU removal, which are collided by DNA replisome during fork reversal. In such scenario, under circumstances of fork reversal by stochastic damage upfront the replisome, PARP1 stacked behind the fork would bump with the reversing replisome inducing PARP1-replication conflicts. In this context, XRCC1 would be required to accelerate PARP1 displacement from the SSB, to promote efficient nick ligation by the POLb-XRCC1-LIG3a complex. This is consistent with the data reported by Ying et al 2016 whereby XRCC1 would facilitate replication fork restart and repair of a subset of unresected or difficult-to-resect stalled replication forks mediated by DNA-PKcs and PARP1.

2) Information is missing in the methods section:

  • Generation and culture of RFP-tagged XRCC1 U2OS cells
  • Immunofluorescence protocol is not described from the blocking, and more importantly, how the fluorescence signals are quantified is not explained.

The methods including the requested information have now been described more thoroughly

3) Last sentences of the different paragraphs of the result section represent interpretations that are not totally supported by the data in the sense that more direct evidences would be necessary, and might therefore be partly integrated in the discussion section.

We thank Reviewer 3 for this helpful comment. We have moved the last sentences of paragraphs from results to discussion sections

4) Figure 1C-D: it could be interesting to show the proportion of positive cells including co-staining. Moreover, use of an S-phase marker or a sensor of arrested replication forks (like FANCD2) would be relevant to support the conclusions.

We have carried out this experiment and quantitation is shown. As 5hmC/5hmU are 2´-deoxynucleosides taken up by cells and incorporate during DNA synthesis, we have also determined the proportion of PAR+-XRCC1+ and g-H2AX+-XRCC1+ co-stained cells upon overnight treatments. We observe a 38% and 52% co-staining of PAR+-XRCC1+ and g-H2AX+-XRCC1+ respectively upon 5hmC exposure (56% of PAR+-XRCC1+ and 52% of g-H2AX+-XRCC1+ co-staining upon 5fC exposure). We have also included a piece of data showing EdU+-XRCC1+ co-staining cells upon 6h and overnight treatments (40% and 28% respectively for 5hmC; 46% and 26% respectively for 5fC). These data suggest to us that XRCC1, PAR and g-H2AX signals most likely associate in S-phase during replication fork progression upon 5hmC and 5fC misincorporation.

5) Chromatin retention assays and immunoblotting Figures 3 and 4: the soluble fraction has to be included. The amount of PARP1 at chromatin cannot be appreciated with H3 blotting but require to show unbound PARP1.

We agree with R3 in this issue, as unbound PARP1 reliably reflects PARP1 dynamics. Soluble fractions have now been SDS-PAGE run and blotted.

6) Figure 5A and B: Authors conclude that 5hmU induces more gH2AX and replication fork arrest than 5hmC (see result section). However, this is not clear regarding the dot plots, and there are no statistical analyzes in this way to support these conclusions.

We conclude that 5hmU induces more DNA damage than 5hmC exposure mainly based on the data presented in Figure 4B where we show a direct comparison of 5hmC and 5hmU (5mM), whereas Figure 5A represents data of cells exposed to low dose 5hmC or 5hmU (1.25mM, at which we do not observe any increase of g-H2AX) in combination with Olaparib. A similar situation happens with the fiber assay described in figure 5B, where we used a low dose (10mM) of 5hmC and 5hmU (which do not induce replication fork defects by their own, to study the additivity with olaparib).

7) Lot of information is missing in the figure legends, especially for drugs concentrations with empty spaces before "M". What is 5C Fig 2B (graph and figure legend)? Nothing about Fig. 4D? Authors should carrefully proofread this part.

We have now corrected this concern and we have gone more carefully throughout the text. We thank R2 for his/her helpful comment.

Minor points:

  • Some acronyms must be explained: TET, 5fC, 5caC, TDG
  • Methods: phospho ser139-H2AX
  • Fig 3C: the difference of staining between 5hmC+olap and the other conditions in XRCC1-/- cells is huge in the picture compared to quantifications in the dot plot.
  • Discussion: it is written that "In agreement with our data, a recently published work shows that contamination of the nucleotide pool by 5hmU-derived from 5hmC leads to 5hmU mis- incorporation in the genome, requiring some components of the BER pathway (SMUG1, PARP1, PARG) to promote cell survival". However, this work (Fugger 2021) focus on 5hmU already present in the chromatin and not newly incorporated.

We again thank to R2 for these comments, which have improved the quality of our MS. Fugger et al. started their work by performing a genome wide Olaparib CRISPR dropout screen in Mus81 deficient cells and found out that DNPH1 and SMUG1, in addition to the FA, BER and related DNA repair pathways, play crucial roles to maintain genome stability. Then, they carried out a second CRISPR screen on cells exposed to 5hmC, confirming that 5hmC is converted to 5hmU by cytoplasmic deaminases, and subsequently incorporated into the genome. Thus, their experimental settings are quite similar to our conditions. But still, the main substantial difference is the replication fork defects that we observe during short exposure, which has important implications for the model of fork protection by XRCC1, as suggested by R2.

We acknowledge R2 for all his/her valuable comments and we hope that we have convinced him/her to recommend publication of our work at IJMS.

Reviewer 3 Report

In this manuscript the authors have investigated the role of XRCC1 in protecting genome from the deleterious effects of various cytosine analogues. Most of the experiments were designed and performed elegantly, however, I have summarized below my major concerns.

  • Among the various cytosine analogues, 5hmC is the most well studied so far and has been suggested to be rather a stable epigenetic mark instead of a transient intermediate. It is therefore quite surprising that exposure of cells to 5hmC leads to such an elevated stress response in the cells. One begs this question how likely it is under physiological conditions that 5hmC levels would lead to such a heightened stress response as the authors have shown in their results. Does this mean that the system used by the authors is a very artificial system where they stressed cells with relatively higher amounts of 5hmC than is usually present within the cell? What is the physiological relevance of this study?

  • The percentage of 5hmC has been shown to be high in brain, liver, kidney and colorectal tissues. Do the authors think that these tissues have higher stress levels owing to higher levels of 5hmC? Does it mean that these tissues have higher levels of genome instability?

  • If 5hmC but not 5mC leads to higher stress response, does it mean that over expression of TET enzymes will also lead to increased stress by increasing the levels of 5hmC and downstream products? It seems like by providing the cytosine analogues, the system is pushed to produce more of the downstream products making cells more sensitive to XRCC1 and BER repair mechanism.

  • In figure 1 and 2 authors have usedγH2AX as a marker of DNA DSBs, however, there are more specific ways to look at the DSBs, such as Comet assay and Pulse Field Gel Electrophoresis.

  • In figure 3, authors have shown replication fork instability after treatment with 5hmC. Although the data does look convincing that after addition of 5hmC in XRCC null cells, the replication fork is stalled (judging by the IdU/CldU ratios). It does seem odd that the WT cells do not show any signs of stalling after 5hmC treatment although PARP1 is significantly accumulated in WT cells as well. It would be nice to perform a supplementary experiment to confirm fork degradation as a result of 5hmC treatment by giving CldU and IdU pulses and adding 5hmC immediately after the second pulse.

  • In several panels authors have shown a chromatin WB for PARP1 and concluded that PARP1 is enriched at the replication forks, however, this is an over-simplification. If the authors wish to investigate the levels of PARP1 at the replication forks, they should perform iPOND assay followed by a WB for PARP1, which would give a precise localization of the proteins at replication forks.

  • The quality of immunofluorescent images needs improvement. Increasing the size of all IF images would serve the purpose. In general, all figures (including the adjoining text) are blurred. Authors should rectify this effect.

  • Please provide images in the supplementary figure for XRCC1-RFP cell line system showing the nuclear and cytoplasmic localization of the over expressed protein.

Author Response

Reviewer 3:

In this manuscript the authors have investigated the role of XRCC1 in protecting genome from the deleterious effects of various cytosine analogues. Most of the experiments were designed and performed elegantly, however, I have summarized below my major concerns.

We thank R3 for his/her compliments about our work, and for helping us to improve our MS by including in the text some of the aspects related to his/her concerns.

Among the various cytosine analogues, 5hmC is the most well studied so far and has been suggested to be rather a stable epigenetic mark instead of a transient intermediate. It is therefore quite surprising that exposure of cells to 5hmC leads to such an elevated stress response in the cells. One begs this question how likely it is under physiological conditions that 5hmC levels would lead to such a heightened stress response as the authors have shown in their results. Does this mean that the system used by the authors is a very artificial system where they stressed cells with relatively higher amounts of 5hmC than is usually present within the cell? What is the physiological relevance of this study?

We thank R3 for his/her insightful ideas about the biological role of 5hmC/5hmU in the genome. We started out investigating this research topic as to how cells differentiate between DNA bases adducted by genotoxic agents and natural and epigenetically modified bases?? Does it exist a mechanism to distinguish these types of base modifications?? The simplest way to challenge replication fork was by feeding cells with modified nucleosides, which would misincorporate during DNA synthesis. We then chose to test this in cells lacking the key BER factor XRCC1 belonging to the so far only-known DNA repair pathway involved in cytosine demethylation. We found that the well-described cytosine demethylation base intermediate 5fC substrate of BER activated BER in our U2OS system. Strikingly, 5hmC exposure also did so, which came to us as a surprise. We also found that the replication fork protection factor FANCD2 also protected cells against 5hmC, as 5hmC challenged replication fork stability in the absence of FANCD2 (under revision in Cell Death and Disease). During these studies, Fugger et al. Science 2021 published that 5hmC, originated either from breakdown of TET-mediated epigenetically-modified DNA or external sources, will contaminate the nucleotide pool subjected to deamination to 5hmU, and misincorporated in the genome, inducing replication fork instability during replication in the following cell cycle. Both TET-mediated DNA modification salvaged nucleotides or externally taken up impacts on genomic instability, although the relative contribution of each process is not well defined. Our data also establish that 5hmC is probably converted to 5hmU (we know that in FANCD2 cells exposed to exogenous 5hmC, 5hmC-derived 5hmU accumulated in the genome at 2000-fold higher than 5hmC. These data suggest to us that 5hmU misincorporation in Xrcc1 cells most likely accounts for the effects described in the current MS, similar to what we found in Fancd2-/- deficient cells. Our study has crucial physiological implications for the DNA repair and epigenetics fields. Tissues with heightened 5hmC in their genomic DNA will probably depend more actively on sanitizing enzymes (such as the DNPH1) to avoid misincorporation of 5-hydroxymethylated nucleosides in their genome.

The percentage of 5hmC has been shown to be high in brain, liver, kidney and colorectal tissues. Do the authors think that these tissues have higher stress levels owing to higher levels of 5hmC? Does it mean that these tissues have higher levels of genome instability?

As described by Fugger et al. those tissues are more likely to have heightened 5hmC/5hmU nucleotide pools. It would be interesting to study the expression of sanitizing enzymes like DNPH1 in those tissues compared to low-5hmC ones. These studies would have important therapeutic implications, as stated in our study, those tissues would show higher response to PARP1 inhibitors (i.e. Olaparib) as a consequence of heightened 5hmU misincorporation, Moreover, we predict that tumors arising in these tissues would probably show higher levels of replication stress sensitive to ATR or PARP inhibition. It would be interesting to test this hypothesis. We thank R3 for the scientific insights.

If 5hmC but not 5mC leads to higher stress response, does it mean that over expression of TET enzymes will also lead to increased stress by increasing the levels of 5hmC and downstream products? It seems like by providing the cytosine analogues, the system is pushed to produce more of the downstream products making cells more sensitive to XRCC1 and BER repair mechanism.

R3 is absolutely right. We have overexpressed the TET1 catalytic domain (TET1CD) in our cells. We have found that cells overexpressing TET1CD not only show heightened 5hmC but also 5fC (and probably 5caC but the dot blot was inconclusive in this very last case) in their genome. TETCD overexpression impacted on Xrcc1-/- cell proliferation as shown in the clonogenic assay, suggesting that XRCC1/BER not only deals with exogenously incorporated but also with TET-mediated 5hmC. However, it is uncertain whether the growth defect observed in TET1-overexpressing cells is due to heightened 5hmC, 5fC or 5caC.

As these are still quite preliminary data, we prefer not to show them in the current MS as they stand.

In figure 1 and 2 authors have used γH2AX as a marker of DNA DSBs, however, there are more specific ways to look at the DSBs, such as Comet assay and Pulse Field Gel Electrophoresis.

 We agree with R3 for his/her comments. Comet assay is a quite reliable but tricky technique to set up in the lab, although we are currently setting it up, without much success. Unfortunately, PFGE is not available in our lab. We hope that γ-H2AX, together with the chromosome aberration test (which is a direct readout of DSBs-harbouring chromosomes fused by the NHEJ machinery) provides a direct evidence of DSB formation during collapsed replication forks upon 5hmC/5hmU exposure.

In figure 3, authors have shown replication fork instability after treatment with 5hmC. Although the data does look convincing that after addition of 5hmC in XRCC null cells, the replication fork is stalled (judging by the IdU/CldU ratios). It does seem odd that the WT cells do not show any signs of stalling after 5hmC treatment although PARP1 is significantly accumulated in WT cells as well. It would be nice to perform a supplementary experiment to confirm fork degradation as a result of 5hmC treatment by giving CldU and IdU pulses and adding 5hmC immediately after the second pulse.

 We acknowledge R3 for this helpful comment. We agree with him/her in his/her concern. So, we do not observe a replication blockage in wildtype cells upon 30´exposure at up to 40mM 5hmC (new Figure 3A), but PARP1 is retained on chromatin upon 3h 5hmC treatment. We have now performed a PARP1 chromatin retention assay upon 5hmC exposure (40mM) for 15, 30 and 60´ and found very little PARP1 retention on chromatin, and relative larger amounts by 5hmU given at the same dose and time. This data suggests that upon 5hmC exposure PARP1 is retained in chromatin longer in the absence of Xrcc1 at a given dose. Secondly, in the presence of Xrcc1, probably the repair kinetics is so fast that we do not detect chromatin bound PARP1 unless 5hmC is maintained for longer than 60´, as shown in new figure 3C.

 Exposure of wildtype cells to 5hmU shows mild PARP1 chromatin retention after 30´ exposure, but quite moderate in the absence of XRCC1. This suggest to us that in the presence of XRCC1, PARP1 is rapidly remove from the chromatin fraction as a reflexion of accurate 5hmC/5hmU repair, whereas PARP1 persist longer in the absence of XRCC1. Regarding to the DNA resection assay, we have also corroborated our results by including data on fork resection defects in Xrcc1 upon exposure to 5hmC or 5hmU, as suggested by R3 (new Figure 3B and 4E), which shows that 5hmC/5hmU exposure increases fork resection of XRCC1 defective forks compared to wildtype. We interpret these data as that in the absence of XRCC1, persistent/trapped PARP1 at 5hmC/5hmU-processing SSBs in nascent DNA leads to fork collapse and extensive degradation

In several panels authors have shown a chromatin WB for PARP1 and concluded that PARP1 is enriched at the replication forks, however, this is an over-simplification. If the authors wish to investigate the levels of PARP1 at the replication forks, they should perform iPOND assay followed by a WB for PARP1, which would give a precise localization of the proteins at replication forks.

 We thank R3 for this his/her valuable comment. We have carried out iPOND assay in our conditions to examined chromatin retained PARP1 with no success. We are currently developing PLA assay, which requires less number of cells. However, this is way far from being achieved. I apologise for this.

The quality of immunofluorescent images needs improvement. Increasing the size of all IF images would serve the purpose. In general, all figures (including the adjoining text) are blurred. Authors should rectify this effect.

 I apologize for this issue again. We uploaded a full MS with High-res images. During editorial processing there must have been a formatting issue affecting image quality. We have now corrected this issue and uploaded full resolution images. 

Please provide images in the supplementary figure for XRCC1-RFP cell line system showing the nuclear and cytoplasmic localization of the over expressed protein.

We have included this new information as part of new Figure 1B.

Reviewer 4 Report

In this manuscript the authors address the role of Xrcc1, a base excision repair (BER) factor, in the misincorporation of cytidine analogous (5hmC and 5hmU ). While the topic may attract the attention of many readers, as it is, the manuscript does not seem to support authors’ claims. Activation of the BER pathway seems convincing in the U2OS cells when 5hmC or 5fC is used, however the effect on fork instability is hard to judge.

In general, quantification is not clear or non-existent (western blots). Fluorescence analysis is not described in the methods. Election of parametric versus parametric statistical test is not justified. Without this information is difficult to agree or disagree with the authors.

Figures require extensive editing: lettering and panels are too small, there is too much white space, panel in figure D is not explained in the legend, units are missing Greek symbols (fig3 says 0.5 M or 10 M?). Furthermore, is there enough precision to say that the concentration of 5hmC is 0.3125  M?

In the absence of line numbers is difficult to provide specific feedback to the authors.

Author Response

Reviewer 4:

In this manuscript the authors address the role of Xrcc1, a base excision repair (BER) factor, in the misincorporation of cytidine analogous (5hmC and 5hmU ). While the topic may attract the attention of many readers, as it is, the manuscript does not seem to support authors’ claims. Activation of the BER pathway seems convincing in the U2OS cells when 5hmC or 5fC is used, however the effect on fork instability is hard to judge.

We thank R4 for his/her comments. We have now incorporated recent data to corroborate the replication fork instability caused by 5-hydroxymethylated nucleosides in the absence of XRCC1 due to trapped PARP1.

In general, quantification is not clear or non-existent (western blots). Fluorescence analysis is not described in the methods. Election of parametric versus parametric statistical test is not justified. Without this information is difficult to agree or disagree with the authors.

We have now provided quantitation and described fluorescence analysis in the methods section. We elected non-parametric test based on the assumption of unknown distribution of the data, with no prior knowledge of fitness to a normal distribution. We used Mann-Whitney test to compared median values of distribution of two populations of cells to ascertain statistical significance.

Figures require extensive editing: lettering and panels are too small, there is too much white space, panel in figure D is not explained in the legend, units are missing Greek symbols (fig3 says 0.5 M or 10 M?). Furthermore, is there enough precision to say that the concentration of 5hmC is 0.3125  M?

 I apologize for this issue about poor quality of images and text. We uploaded a full MS with High-res images, but I believe that there must have been a formatting issue affecting image quality during editorial processing. We have now corrected this issue and have uploaded full resolution set of images. We have now corrected 5hmC concentration to 0,3mM.

In the absence of line numbers is difficult to provide specific feedback to the authors.

I apologize for this issue again. We hope that the new version of the MS is more suitable for R4´s comments and criticisms. 

Round 2

Reviewer 1 Report

Major improvements have been made by the authors and several aspects are now clarified.

However, a few more things need to be addressed before this is being accepted for publication.

- A representative illustration of the proposed mechanism highlighting the Parp1-trapped DNA lesions role in the genomic instability is still missing, even if a detailed explanation is now added in the manuscript. The latter, should be provided in a comparative manner, in relation to Fugger et al. work.

- Most of the Figures’ quality is low, thus important information can be easily missed out by the reader (i.e. Figure 5C).

- I was not able to locate the addition made in the manuscript regarding the reply provided on the following previous comment: “The authors should provide an explanation on their strategy on selection of the specific 2’-deoxycytidine derivatives/lesions for their studies.“

Author Response

Reviewer 1:

Major improvements have been made by the authors and several aspects are now clarified.

However, a few more things need to be addressed before this is being accepted for publication.

- A representative illustration of the proposed mechanism highlighting the Parp1-trapped DNA lesions role in the genomic instability is still missing, even if a detailed explanation is now added in the manuscript. The latter, should be provided in a comparative manner, in relation to Fugger et al. Work.

We thank reviewer 1 for his/her comments. We have now included a model depicting the alternative situations in the absence/presence of XRCC1, including the situation described by Fugger et al. We hope that R1 agrees qith our proposed model of genomic instability

- Most of the Figures’ quality is low, thus important information can be easily missed out by the reader (i.e. Figure 5C).

We have included high resolution figures, in addition to the ones provided in the MS template. We hope that MDPI publishing group uses the original images to solve this issue.

- I was not able to locate the addition made in the manuscript regarding the reply provided on the following previous comment: “The authors should provide an explanation on their strategy on selection of the specific 2’-deoxycytidine derivatives/lesions for their studies.“

As TDG during BER has been found to play a crucial role during removal of 5fC and 5caC from template DNA, we started this project thinking that cells lacking the BER factor XRCC1 would be sensitive to misincorporation of these cytidine analogues exclusively. To our surprise we found XRCC1 deficient cells to be hypersensitive to 5hmC and to 5fC to a lesser extent. This was the rational behind the work presented here. Because 5hmC can be deaminated in vivo to 5hmU (a nucleoside with a similar chemical structure to 5hmC), we also tested the sensitivity to 5hmU and found that 5hmU presented a similar behavior to 5hmC. During the development of this work, Fugger et al confirmed that 5hmU originated from 5hmC.

We included in the previous version of the MS the following paragraph the addition:

 Whereas the role of BER in removing epigenetically modified cytidine analogues from the template DNA is well understood, little is known about its role during misincorporated misincorporation of 2-deoxycytidine analogues 5mC, 5hmC, 5fC and 5caC during replication is currently unknown.”

We hope this sentence provides an explanation for the selection of the specific strategy used in this study.

Reviewer 3 Report

The revised version has satisfactorily addressed my concerns. 

Author Response

We acknowledge R3 for his/her valuable comments. Thank you.